# A scabies outbreak in the North East Region of Ghana: The necessity for prompt intervention

Yaw Ampem Amoako[1,2]*, Richard Odame Phillips[2,3], Joshua Arthur[1], Mark Ayaaba Abugri[4], Emmanuel Akowuah[2], Kwabena Oppong Amoako[2], Benjamin Aboagye Marfo[5], Michael Frimpong[2], Tjip van der Werf[6], Sofanne Jacobine Ravensbergen[6], Ymkje Stienstra[6]

1 Komfo Anokye Teaching Hospital, Kumasi, Ghana, 2 Kumasi Center for Collaborative Research in Tropical Medicine, Kwame Nkrumah University of Science and Technology, Kumasi, Ghana, 3 School of Medicine and Dentistry, Kwame Nkrumah University of Science and Technology, Kumasi, Ghana, 4 East Mamprusi District Health Directorate, Gambaga, Ghana, 5 Neglected Tropical Diseases Control Programme, Ghana Health Service, Accra, Ghana, 6 University Medical Center, Groningen, Netherlands

* yamoako2002@gmail.com

**Data Availability Statement:** All relevant data are within the manuscript and its Supporting Information files.

## Abstract

### Background

There is a dearth of data on scabies from Ghana. In September 2019, local health authorities in the East Mamprusi district of northern Ghana received reports of scabies from many parts of the district. Due to on-going reports of more cases, an assessment team visited the communities to assess the effect of the earlier individual treatment on the outbreak. The assessment team furthermore aimed to contribute to the data on scabies burden in Ghana and to demonstrate the use of the International Alliance for the Control of Scabies (IACS) diagnostic tool in a field survey in a resource limited setting.

### Methodology/Principal findings

This was a cross sectional study. Demographic information and medical history was collected on all participants using a REDCap questionnaire. A standardised skin examination of exposed regions of the body was performed on all participants. Scabies was diagnosed based on the criteria of the International Alliance for the Control of Scabies (IACS). Participants were mostly female (61.5%) and had a median age of 18.8 years (IQR 13–25). Two hundred out of 283 (71%) of participants had scabies with most (47%) presenting with moderate disease. Impetigo was found in 22% of participants with scabies and 10.8% of those without scabies [RR 2.27 (95% CI 1.21–4.27)]. 119 participants who received scabies treatment in the past months still had clinical evidence of the disease. 97% of participants reported a recent scabies contact. Scabies was commoner in participants $\leq$16 years compared to those >16 years [RR 3.06 (95% CI 1.73–5.45)].

### Conclusion/Significance

The prevalence of scabies was extremely high. The lack of a systematic approach to scabies treatment led to recurrence and ongoing community spread. The IACS criteria was

**Funding:** YS received financial support from Stichting Buruli ulcer Foundation (https://buruli1ulcer2groningen3.wordpress.com/) and the Gratama Foundation (grant number 2017-029 https://www.rug.nl/alumni/support-research-and-education/groninger-university-fund/gratama-stichting/). The funders had no role in study design, data collection and analysis, decision to publish, or preparation of the manuscript.

**Competing interests:** The authors have declared that no competing interests exist.

useful in this outbreak assessment in Ghana. Alternative strategies such as Mass drug administration may be required to contain outbreaks early in such settings.

## Author summary

Scabies, recently categorised as a Neglected Tropical Disease by the WHO is caused by infestation with *Sarcoptes scabiei* and is characterised by intense pruritus and rash that typically involves the genitalia and the web spaces of the fingers and toes. It has a large global burden and is associated with significant morbidity and socio-economic burden. Secondary bacterial infections following scabies can lead to significant complications including chronic kidney disease from glomerulonephritis and possibly rheumatic heart disease. An outbreak of scabies was reported in Ghana's East Mamprusi district in September 2019. Despite earlier treatment of individual cases, scabies prevalence was 71%. About 19% of participants had impetigo which was mostly mild in severity. Absence of a systematic approach to treat scabies led to recurrence and ongoing community spread. The recently published IACS criteria for diagnosing scabies proved useful in this outbreak assessment in Ghana. Alternative strategies such as Mass drug administration may be required to contain outbreaks in such settings.

## Introduction

Scabies is an intensely pruritic skin disease caused by the mite *Sarcoptes scabiei* and significantly impacts the quality of life of affected persons [1]. The Global Burden of Diseases study estimates that scabies affects 455 million persons leading to approximately 3.8 million disability-adjusted life-years (DALYs); making scabies one of the most common neglected tropical diseases [2]. Recently, the WHO has classified scabies as a Neglected Tropical Disease (NTD) to highlight the need for prioritisation of the condition in low and middle income countries [1].

Scabies and its associated acute symptoms and secondary complications pose a significant socioeconomic burden to affected persons, families, communities and the health system. Direct financial costs of scabies relate to the cost of medicines, loss of productivity, and institutional outbreaks resulting from hospitalization of cases [1]. The intense pruritus (during the initial illness or the post scabies itch) severely affects sleep, work, and the quality of life [3]. Scabetic lesions and the excoriations following skin scratching can result in superinfection with bacteria (commonly, *Streptococcus pyogenes* and *Staphylococcus aureus*) that can result in complications such as pyoderma, impetigo, and cellulitis. Infection with *S. pyogenes* can also lead to immune-mediated complications including post streptococcal glomerulonephritis and possibly acute rheumatic fever, which can further lead to chronic kidney disease and rheumatic heart disease respectively [1,4].

Scabies transmission is predominantly by direct contact (including sexual contact) with infected skin [5,6]. Although less common, contact with infested fomites including clothing, bedding and towels have been thought to play a role [5,7]. Outbreaks of scabies have occurred during wars, refugees/migration crisis [8] and in situations of overcrowding such as in schools [9–11], prisons and care homes [12]. Scabies has been reported to be associated with household size, low socioeconomic status and poor access to healthcare [13–15].

A systematic review conducted in 2015 estimated scabies prevalence worldwide to range from 0.2% to 71.4% depending on the populations studied [16]. The greatest burden of scabies is in low- and middle-income countries where overcrowding and inadequate access to effective treatment serve as drivers of disease transmission.

Two hospital-based studies conducted in Accra and Kumasi, the two largest cities of Ghana, reported scabies rates of 5.1% and 12.4% respectively [17,18]. In September 2019, local authorities received reports of scabies among communities in the East Mamprusi district in the North East Region of Ghana The district health team treated individual patients with scabies within the communities with topical benzyl benzoate. Due to continued reports of scabies, our medical team visited the district more than 3 weeks after individual treatment efforts by the local health authorities to further assess the scabies burden. This was an opportunistic assessment that was undertaken in the context of an outbreak investigation in Northern Ghana. The assessment team aimed to study the outbreak to contribute to the data on scabies burden in Ghana and to demonstrate the use of the International Alliance for the Control of Scabies (IACS) diagnostic tool in a field survey in a resource limited setting. In this study we report the use of the IACS diagnostic tool which allowed for systematic evaluation of scabies in a field survey in rural Ghana.

## Methods

### Ethics statement

All participants provided written informed consent. Written permission was also obtained from the district health authorities. For young children within the communities, written consent was obtained from parents or legal guardians. In the school, in addition to obtaining verbal permission from the school authorities which informed parents about the research activities, school children age < 18 years provided verbal assent and verbal consent was also obtained from their parents or legal guardians. Ethical approval for the study was granted by the Committee on Human Research, Publications and Ethics (CHRPE) of the School of Medical Sciences of the Kwame Nkrumah University of Science and Technology (approval number: CHRPE/AP/671/19) in Ghana and the University Medical Center Groningen Institutional Review Board (approval number 201900650) in the Netherlands.

### Study procedures

This was a cross-sectional study conducted in the East Mamprusi district which is located in the recently created North-East region of Ghana. Dwellings in this region are typically round mud houses with thatch roofs; although the houses have variable sizes, most households have 2–3 rooms where all inhabitants of the household sleep.

Community members and senior high school students were invited to participate in an interview during house to house visits and a school visit. The assessment team started their activities in the communities and a boarding school where local health authorities reported the burden to be high. The assessment team consisted of medical doctors with clinical experience diagnosing scabies based on earlier activities in infectious diseases and/or public health. In addition, a supplemental training program on the diagnosis of scabies, impetigo and other locally common skin conditions as well as the use of the IACS criteria for scabies diagnosis was provided to assessment team members. In the school, there was a random selection of student participants; per classroom, one tenth of the students were invited to participate based on students' seating in the classroom. Per community, a house to house visit was performed. All persons present in the house at the time of the visit were invited to participate. Five different communities were visited.

Basic demographic information of the participants were recorded using a REDCap based questionnaire (S1 Text) which was hosted in a database located at the University Medical Center, Groningen, Netherlands. A medical history was followed by a standardised skin examination of the exposed regions of the skin as was done in a previous study from Solomon Islands [10]. Briefly skin examination consisted of assessment of exposed areas: the feet and legs to the thighs, hands to the upper arms, neck, face and scalp. Students were in school uniform which consisted of above-knee shorts and above-elbow shirts or dresses. Shoes were removed prior to examination. Adults were also required to have the designated body regions exposed prior to their skin examination. The examination excluded breasts and genitals, unless requested by participants and then only in a separate, private examination area. A focused history of standardized questions was taken of all participants consisting of information required for the IACS criteria classification. Questions included whether participants experienced itch. Contact history was assessed by asking if participants lived with someone, or had a friend or classmate with itch, or if they lived with someone, or had a friend or classmate with a rash that looks like scabies. Participants were shown images of people with typical scabies rashes to assist these questions. Questions on treatment included whether participants had received any scabies treatment in the preceding two months, what treatment was received if any and a description of how the treatment was used. History was taken in the local language (with the assistance of interpreters where required) or in English.

The diagnosis of scabies was based on the B1, B3, C1 and/or C2 criteria developed by the IACS as shown in Table 1 [19]. The assessment team also looked for crusted scabies.

Impetigo was diagnosed based on the presence of papules, pustules or ulcerative lesions with associated erythema, crusting or pus. The severity of scabies and impetigo were assessed using previously published criteria [20] based on the number of lesions present. Scabies was categorized as: mild, 1 to 10 lesions; moderate, 11 to 49 lesions; or severe, 50 or more lesions. Impetigo was classified as: very mild, 1 to 5 lesions; mild, 6 to 10 lesions; moderate, 11 to 49 lesions; or severe, 50 or more lesions. Benzyl benzoate was supplied to participants with scabies and their contacts as per standard protocol in Ghana [21]. Participants with impetigo were also treated as per standard protocol in Ghana [21].

**Table 1. Case definitions for scabies using the IACS criteria.**

| Clinical category | | Used in survey |
|---|---|---|
| Confirmed scabies | | |
| A1 | Mites, eggs or faeces on light microscopy of skin samples | No |
| A2 | Mites, eggs or faeces visualised on individual using high powered imaging device | No |
| A3 | Mite visualised on individual using dermoscopy | No |
| Clinical scabies | | |
| *B1 | Presence of burrows | Yes |
| B2 | Typical lesions affecting male genitalia | No |
| B3 | Typical lesions in a typical distribution and two history features (itch and contact history) | Yes |
| Suspected scabies | | |
| C1 | Typical lesions in a typical distribution and one history feature (itch or contact history) | Yes |
| C2 | Atypical lesions or atypical distribution and two history features (itch and close contact with an individual who has itch or typical scabies lesions in a typical distribution) | Yes |

*Burrows were not confirmed with dermoscopy in the study

## Statistical analysis

We conducted descriptive and inferential statistical analyses to present the data of the outbreak. Categorical variables were expressed as frequencies and proportions; and results for continuous variables were expressed as median and interquartile range (IQR). The severity of scabies and impetigo in the earlier treated and untreated groups were compared using the Mann-Whitney test. The Relative Risk (RR) of impetigo in participants with or without scabies was calculated with 95% confidence interval (CI). A p value <0.05 was set as the level of statistical significance. Statistical analysis was performed using IBM SPSS statistics Version 20 (IBM Company, Armonk, NY, USA).

## Results

In total, 283 participants were interviewed. No one refused participation. Ninety three students (of 448 in session) and 5 different communities were visited, including 190 participants in the house-to-house visit. The majority were female (61%) and the median age of the participants was 19 (IQR 13–25) years (Table 2). The most frequently reported occupation was farming (30%).

Based on the IACS criteria, 71% of the 283 participants were diagnosed with scabies (Table 3). Skin examination revealed burrows in 37.0% and rash typical for scabies in 97% of the scabies cases. Most participants with scabies had moderate disease. Scabies lesions were mostly located on hands, fingers and finger webs (Fig 1). No cases of crusted scabies were observed. Fifty three participants had impetigo of varying severity.

Compared to participants previously treated with benzyl benzoate, scabies burrows were more prevalent among untreated participants. In a post-hoc analysis, there was a statistically significant difference in B3 and B1 diagnostic classification between the participants who were treated with benzyl benzoate in the past months and untreated participants (p < 0.05). Itch was reported by 79% participants with a median duration of 30 days (IQR 21–60). Only 3% of the participants had no previous known scabies contact in the past weeks. At the time of the interview, 59.4% of the 200 participants with scabies had recently received treatment with topical benzyl benzoate (first line treatment in Ghana) in the past two months because of their skin problems. 117 of these 119 previously treated did not only have an itch but also demonstrated skin manifestations which lead to an IACS scabies diagnosis. No one received

**Table 2. Baseline characteristics of participants.**

|  | Number n = 283 |
|---|---|
| Female (%) | 174 (61.5) |
| Missing information (%) | 4 (1.4) |
| Age median (IQR) | 18.8 (13.0–25.0) |
| Missing information (%) | 3 (1.0) |
| **Education/work** | |
| Preschool (%) | 11 (3.9) |
| Primary school (%) | 26 (9.2) |
| Junior high school (%) | 2 (0.7) |
| Senior high school (%) | 94 (33.2) |
| Farmer (%) | 86 (30.3) |
| Other*(%) | 50 (17.7) |
| Missing information (%) | 14 (4.9) |

*Other includes occupations such as trading and sewing

**Table 3. Scabies and impetigo severity in participants with or without clinical scabies.**

| | | Clinical scabies, not treated | Clinical scabies, previously treated | No clinical scabies, not treated | No clinical scabies, previously treated |
|---|---|---|---|---|---|
| **Total (n = 283)** | | 81 | 119 | 68 | 15 |
| **IACS category** | B1 (%) | 49 (60.5) | 25 (21.0) | NA | NA |
| | B3 (%) | 23 (28.4) | 84 (70.6) | | |
| | C1 (%) | 4 (4.9) | 7 (5.9) | | |
| | C2 (%) | 5 (6.2) | 3 (2.5) | | |
| Positive contact history (%) | | 81 (100) | 118 (99.2) | 61 | 15 |
| **Scabies severity** | | | | | |
| Mild (%) | | 34 (42.0) | 37 (31.1) | NA | NA |
| Moderate (%) | | 38 (46.9) | 56 (47.1) | | |
| Severe (%) | | 9 (11.1) | 23 (19.3) | | |
| Missing information (%) | | 0 (0.0) | 3 (2.5) | | |
| **Impetigo severity** | | | | | |
| Very mild (%) | | 10 (12) | 6 (5) | 2 (3) | 1 (7) |
| Mild (%) | | 9 (11) | 9 (8) | 3 (4) | 1 (7) |
| Moderate (%) | | 3 (4) | 7 (6) | 2 (3) | 0 (0) |
| Severe (%) | | 0 (0) | 0 (0) | 0 (0) | 0 (0) |

NA = not applicable

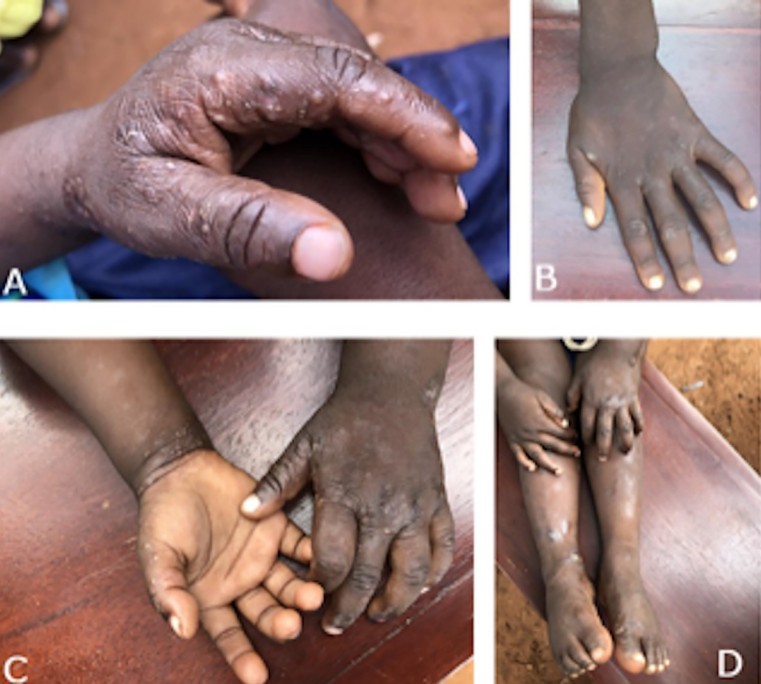

**Fig 1. Scabies lesions seen on exposed body regions in four selected participants**

**Table 4. Skin problems and contact history in participants with and without scabies.**

|  | Participants with scabies, n = 200 (%) | Participants without scabies, n = 83 (%) | All participants n = 283 (%) |
|---|---|---|---|
| Itch present | 190 (95.0) | 34 (40.9) | 224 (79.2) |
| Rash present | 191 (95.5) | 22 (26.5) | 213 (75.3) |
| *Contact history positive | 199 (99.5) | 76 (91.6) | 275 (97.2) |

*Contact history is considered positive if school or house contact with itch or rash was reported

permethrin or ivermectin. Only nine persons with scabies reported using herbal medicine as treatment for their scabies. There was no statistically significant difference in the severity of scabies between the participants, who were treated with benzyl benzoate in the past months and the untreated participants (p = 0.068). Only 15 of the participants previously treated for scabies were free of the disease at the time of the interview. Even in participants without clinically evident scabies, itch and rash were common (40.9% and 26.5% respectively) but this was much less than in those with scabies (95% and 95.5% respectively) as shown in Table 4. Impetigo was found in 22% of participants with scabies and 10.8% of those without scabies [RR 2.27 (95% CI 1.21–4.27)]. One hundred and nineteen (119) participants who had been previously treated still had clinical evidence of scabies. Scabies was commoner in participants ≤16 years compared to those >16 years [RR 3.06 (95% CI 1.73–5.45)].

## Discussion and conclusion

The scabies prevalence worldwide varies widely and depending on the population studied ranges from 0.2% to 87% [16,22] with the highest prevalence in the island nations of the Pacific and countries in Latin America. Scabies is reported to be more prevalent in children than in adolescents and adults. In a national survey in Fiji, the overall prevalence of scabies was found to be 23.6% with rates of 43.7% and 36.5% in children aged 5–9 years and <5 years respectively [23]. The prevalence of 71% reported in the present study is extremely high but lower than the 87% reported previously from a village in Papua New Guinea [22]. It is however higher than the 32% reported from Fiji by Haar and colleagues [24]. A scabies rate of 17.6% with peak infection occurring in children 5–9 years was reported when villagers in a community in the Ashanti region of Ghana were studied over 3 decades ago [25]. More recently, two hospital based studies and one school based study conducted in Accra and Kumasi, the two largest cities of Ghana, reported rates of 5.1%, 12.4% and 11.5% respectively [9,17,18]. A degree of selection bias may have impacted the high prevalence found in the present study as the study was conducted among communities where the local health authorities had reported a high scabies burden at the onset of the outbreak. The much higher prevalence found in our study is probably due to the fact that the population of East Mamprusi is predominantly rural, have larger-sized households and higher level of poverty compared to especially, southern parts of Ghana. Such an extremely high prevalence of scabies as reported in this study has its impact on the communities' quality of life and likely results in loss of productivity at school and work.

The relative risk of impetigo in participants with scabies was 2.27 (95% CI 1.21–4.27). Eighty three (83%) of participants with impetigo also had scabies. This is similar to findings from the Solomon Islands where 63.5% of those with impetigo had scabies [10]. These findings have implications for the control of scabies and impetigo in endemic populations like described in the current study. Indeed, mass drug administration (MDA) with ivermectin has been reported to result in a decrease in prevalence of scabies with an added benefit of about 90% relative reduction in the prevalence of impetigo [26].

The IACS criteria of scabies consist of three diagnostic categories (confirmed, clinical and/ or suspected scabies) [19]. The clinical and suspected categories are practical for use, help to standardize reporting on scabies, and can easily be applied in field surveys like ours. Scabies and impetigo infections are reported to be under-recognised and hence under-treated by clinicians [1,27] as laboratory diagnosis is impossible in most settings [1]. The use of the IACS criteria for diagnosis may result in improvements in the recognition and treatment of scabies infections by clinicians. The IACS criteria proved its utility when used for a school survey in the Solomon Islands [10]. The current study has further demonstrated the use of the IACS diagnostic criteria to systematically evaluate for scabies in an outbreak setting in a rural area in Sub-Saharan Africa.

Scabies affected predominantly young persons in this study. Most patients with scabies were aged ≤16 years. This is similar to the epidemiology of other NTDs of the skin like Buruli ulcer [28,29] and Yaws [30]. This provides an opportunity for integrating detection and control activities of these skin NTDs using school-based programmes [28,31–33].

Treatment options for scabies in Ghana include 5% permethrin and topical benzyl benzoate which is usually available as a 25% formulation. Permethrin is not widely available and is relatively unaffordable to the rural population which are most affected by scabies. Benzyl benzoate is relatively more readily available and cheaper. At the present time, ivermectin is not licensed for scabies treatment in Ghana (an exception may be in crisis situations where the Ghana Health Service may grant an emergency use authorisation).

Benzyl benzoate was used for the treatment of individual cases in keeping with standard practice as recommended by Ghana's standard treatment guidelines. The treatment with benzyl benzoate of the individual scabies cases in the different communities proved insufficiently effective to control the outbreak. One possible explanation could be that benzyl benzoate is less effective for the treatment of scabies compared to other treatment options like ivermectin or permethrin [6]. Benzyl benzoate administered topically as a 25% solution may cause skin irritation especially in younger children and this may negatively impact compliance with therapy. In Ghana, the standard treatment guidelines requires application of benzyl benzoate over the whole body (except the face) twice and left overnight on two consecutive nights. The first application is done after a warm bath with the application repeated the next day (without a bath) and washed off 24 hours later. It is plausible that persons receiving the treatment did not fully adhere to the instructions on the use of benzyl benzoate. Contacts of cases are usually advised to treat themselves at the same time as the case in order to reduce the risk of re-infection [5,34–36]. The lack of treatment of household contacts is potentially a major factor leading to re-infection of cases and ongoing community transmission. This is further supported by the fact that 119 participants who were treated still met the IACS criteria for scabies after treatment. Only 9% of the participants who were previously treated with benzyl benzoate had no scabies at the time of the interviews. The limited access to health care as well as the relatively high number of residents per household in the current outbreak zone presumably drive ongoing spread and/or re-infections.

The interval from scabies treatment to response is variable. The rash and itch may persist for up to 4 weeks after treatment. Patients treated in the preceding 2 weeks might still have symptoms thus making it difficult to distinguish re-infections from primary treatment failure [5,7,35]. It is possible that some previously treated patients with an IACS classification of B3 may have resolving scabies rather than re-infection; the lower prevalence of burrows in the treated group compared with those not treated would support this possibility. However, given the timing of reassessment 3 weeks to 2 months after treatment, resolution of symptoms in the majority of treated patients would be expected [37].

Some participants reported itch and rash but had no scabies as defined by the IACS criteria. This group of participants probably consists of persons with alternate aetiologies of itch and rash such as fungal skin infections. Yet some in this group may be developing scabies considering the percentage who had a positive contact history. Additionally, some may represent persons with post scabies itch.

A control strategy of treating clinical cases and their contacts undoubtedly provides relief for individuals with scabies, but its success in reducing population prevalence in the longer term is limited [34]. MDA using topical permethrin or oral ivermectin offers an alternative approach for population control to substantially reduce the burden of scabies. Among populations of northern Australia [38,39], Egypt [15] and in Panama [40], mass treatment of highly endemic communities with topical 5% permethrin substantially reduced scabies prevalence. In another study with permethrin, although the scabies prevalence remained unchanged, the prevalence of secondary infected scabies decreased from 3.7% to 1.5% representing a relative reduction of 59% [41].

Earlier studies [24,26,42] on MDA with ivermectin were performed in island settings and reduced the prevalence of scabies, even 24 months after the intervention [26]. In a study done in Fiji, ivermectin resulted in a 94% and 89% relative reductions in scabies prevalence at 12 and 24 months respectively; and these reductions in scabies prevalence were greater in the ivermectin group than in persons who received MDA with permethrin [26]. MDA for scabies is indicated if the community prevalence is more than 10% [43]. Subsequently the Neglected Tropical Diseases Programme in Ghana decided to provide MDA with ivermectin for the district described in this report. However, difficulties arise determining the size of the MDA needed to reduce the scabies burden if interactions in a population are not limited based on geography (e.g. islands or rivers) or social factors (e.g. institutions). To control such outbreaks in the future, and to develop a global control programme, studies on optimal implementation of MDA, especially in larger, non-isolated areas are desperately needed [1,26].

## Study limitations and strengths

The sample size of this study is relatively small compared to the population of the entire district. Even though there may be differences in scabies prevalence between communities in the district, due to the close person-to-person interactions between communities (resulting from close extended family ties), it is unlikely that such a percentage would be lower than 10% (MDA indicated if higher) considering the high percentage of scabies found in the communities studied. The examined students come from different communities across the district. Furthermore, the district health team ended up providing MDA across the district and did not observe communities without scabies problems.

## Conclusion

This study provides data on scabies burden and could form the basis for guiding future research in Ghana and West Africa where there is a dearth of data on scabies prevalence. The IACS criteria for standardisation of scabies diagnosis was easily and practically applied in our field survey in a resource limited setting in rural Ghana.

## Supporting information

**S1 Text. Scabies outbreak questionnaire**
(DOCX)

## Acknowledgments

We thank the Informatiemanagement Onderzoek at the University Medical Centre Groningen for the greatly appreciated support with REDCap mobile app. We are grateful to staff of the Northern Regional Health Directorate and the East Mamprusi District Health Directorate for their assistance.

## Author Contributions

**Conceptualization:** Yaw Ampem Amoako, Richard Odame Phillips, Tjip van der Werf, Ymkje Stienstra.

**Data curation:** Yaw Ampem Amoako, Joshua Arthur, Mark Ayaaba Abugri, Emmanuel Akowuah, Kwabena Oppong Amoako, Benjamin Aboagye Marfo, Michael Frimpong.

**Formal analysis:** Yaw Ampem Amoako, Sofanne Jacobine Ravensbergen.

**Funding acquisition:** Richard Odame Phillips.

**Investigation:** Yaw Ampem Amoako, Joshua Arthur, Mark Ayaaba Abugri, Emmanuel Akowuah, Kwabena Oppong Amoako, Benjamin Aboagye Marfo, Michael Frimpong.

**Methodology:** Yaw Ampem Amoako, Richard Odame Phillips, Tjip van der Werf, Sofanne Jacobine Ravensbergen, Ymkje Stienstra.

**Project administration:** Yaw Ampem Amoako, Richard Odame Phillips, Ymkje Stienstra.

**Supervision:** Richard Odame Phillips, Tjip van der Werf, Ymkje Stienstra.

**Validation:** Richard Odame Phillips.

**Writing – original draft:** Yaw Ampem Amoako, Sofanne Jacobine Ravensbergen, Ymkje Stienstra.

**Writing – review & editing:** Yaw Ampem Amoako, Richard Odame Phillips, Joshua Arthur, Mark Ayaaba Abugri, Emmanuel Akowuah, Kwabena Oppong Amoako, Benjamin Aboagye Marfo, Michael Frimpong, Tjip van der Werf, Ymkje Stienstra.

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
