## [Decision Letter · Decision Letter 0]

2 May 2020

Dear Dr Amoako,

Thank you very much for submitting your manuscript "A scabies outbreak in the North East Region of Ghana: the necessity for prompt intervention" for consideration at PLOS Neglected Tropical Diseases. As with all papers reviewed by the journal, your manuscript was reviewed by members of the editorial board and by several independent reviewers. In light of the reviews (below this email), we would like to invite the resubmission of a significantly-revised version that takes into account the reviewers' comments. 

We cannot make any decision about publication until we have seen the revised manuscript and your response to the reviewers' comments. Your revised manuscript is also likely to be sent to reviewers for further evaluation.

Sincerely,

Alberto Novaes Ramos Jr., M.D.

Guest Editor

Aysegul Taylan Ozkan

Deputy Editor

Reviewer's Responses to Questions

**Key Review Criteria Required for Acceptance?**

**Methods**

-Are the objectives of the study clearly articulated with a clear testable hypothesis stated?

-Is the study design appropriate to address the stated objectives?

-Is the population clearly described and appropriate for the hypothesis being tested?

-Is the sample size sufficient to ensure adequate power to address the hypothesis being tested?

-Were correct statistical analysis used to support conclusions?

-Are there concerns about ethical or regulatory requirements being met?

Reviewer #1: Please see Summary and General Comments!

Reviewer #2: (No Response)

Reviewer #3: The population is partially described - if possible the authors should give a estimate of the number of communities and households in the region. 

It is unclear as to how representative the sample of 283 patients is of the whole population (This should at least be acknowledged / or discussed in the discussion)

**Results**

-Does the analysis presented match the analysis plan?

-Are the results clearly and completely presented?

-Are the figures (Tables, Images) of sufficient quality for clarity?

Reviewer #1: Please see Summary and General Comments!

Reviewer #2: (No Response)

Reviewer #3: data analysis mostly matches plan - some comparative statistics provided - no odds ratios provided with p-values, not clear in methods how these comparisons were made

**Conclusions**

-Are the conclusions supported by the data presented?

-Are the limitations of analysis clearly described?

-Do the authors discuss how these data can be helpful to advance our understanding of the topic under study?

-Is public health relevance addressed?

Reviewer #1: Please see Summary and General Comments!

The limitations of analysis are not clearly discribed.

Reviewer #2: (No Response)

Reviewer #3: The conclusions are not fully supported by the data - particularly regarding the need for MDA (which was not feasible in this case) and the reason for ongoing transmission being related to the lack of treatment of contacts.

Limitations and strengths are not discussed

The implications of the findings are not fully explored.

**Editorial and Data Presentation Modifications?**

Reviewer #1: Please see Summary and General Comments!

The topic of the article is highly relevant, but the presentation has too many weaknesses and needs to be rewritten.

Reviewer #2: (No Response)

Reviewer #3: The authors summarise the importance of scabies and the data available to date from Ghana. The use of the IACS criteria is relatively novel and demonstrates the utility of this tool in the field setting. The authors demonstrate a high prevalence of scabies in the setting of an regional ourbreak, adding to limited national data on prevalence of scabies. There is discussion the potential role of MDA and limitations of this approach. 

The methods and results should provide a clearer picture of how representative the sample of 283 participants is if possible. The discussion should be expanded to discuss the use of IACS criteria in the context of previous studies, the limitations of the IACS criteria, the strengths and limitations of the study and implications for future research. The discussion on MDA is of interest but perhaps a more broad discussion of other available strategies, including limitations (particularly as MDA was not feasible in this instance). 

Abstract

- separate methodology and results

- revise statement on mass drug administration - perhaps "maybe an effective strategy in such settings to contain outbreaks"

- line 24 - change "remanifestations" to "recurrence and ongoing community spread"

Author summary

- line 31 - change "huge" to "large"

- line 33 - change "of" to "complicating scabies can lead to significant complicatiosn including..."

- line 38 - change "remanifestations" as above

- line 40 - revise statement on mass drug administration as above

Introduction

- line 69 - reference?

Methods

- what is the estimated population of the area examined? (if available)

- how many communities were examined? what is the estimated size of the school?

- line 101 - dose this mean - all houses in each community were visited and ?all members of the household were invited to participate?

- how was consent obtained for school children? 

- how many team members conducted assessments and was any specific training required to perform IACS assessment?

- include in statistics how comparisons were made (between treated and non-treated patients?) 

Results

- what proportion of the population does the 283 people represent? (appreciate there may not be readily available census data?)

- line 158 - provide odds ratio + 95% CI with P value 0.063

- line 159 - provide odds ratio + 95% CI with P value 0.115

- data on age stratified prevalence would be interesting if available, also gender differences

Discussion

- line 177 - 87% contradicts earlier statement on line 172 (0.2% to 71.4%)

- line 208 - reference? Cochrane review on treatment comparison https://onlinelibrary.wiley.com/doi/abs/10.1002/ebch.861

- may be helpful to give previous figures (percentage response from clinical triasl)

- ?discuss difficulties with compliance

- line 208 - yet "the main reason" should be revised - the data does not completely support this conclusion - something like "the lack of treatment of household contacts is potentially a major factor leading to re-infection of cases and ongoing community transmission."

- line 210 - discuss time to response following treatment - is it possible that some of these patients had 'resolving' scabies (e.g. those treated within 1-2 weeks of assessment) 

- discuss rationale for use of IACS criteria - health care worker recognition of scabies may be poor? https://www.ncbi.nlm.nih.gov/pubmed/28671945 - laboratory diagnosis is impossible in most settings 

- discuss previous studies to report on IACS criteria utility - where does this study sit

- discuss rationale for selecting representative population in schools (vs whole school) 

- what if any treatment was provided to cases / contacts

- discuss limitations and strengths of the study

**Summary and General Comments**

Reviewer #1: Scabies is one of the world’s commonest afflictions and despite that a neglected disease. As such it warrants public health interventions and a global control programme. There are still, however, several key operational research questions to be addressed, e.g. prevalence data, before mass drug interventions could be outlined. The present article dealing with a local scabies outbreak in Ghana should therefore be of great scientific interest and merit publication in PLOS. Of the 11 authors, seven are from Ghana and three from the Netherlands. The affiliation of Michael Frimpong is lacking. 

Unfortunately, I cannot recommend acceptance of the article as it now stands. Generally, it needs more editing, more consideration for details, better structure and focus, less repetitions and better information for the readers to be able judge on the internal and external validity.

The incitement for the data collection and the study was a reported outbreak in northern Ghana communities. Despite individual treatments the transmission of scabies had not been appreciably reduced. It would be interesting to know whether Ghana has standardised instructions for individual case-management of scabies and impetigo.

Some specific comments

The Introduction should give the readers information on scabies generally and then zoom in on the general situation in Ghana. We need to understand a bit about how the health care operates and what is meant with a district, a municipality, a community, a village. A figure could be of help.

The introduction should end with clear stated aims. “In this study we report ............” is all too vague. Using a descriptive epidemiological Methodology the authors report prevalence data and assess the impact of earlier individual treatment on their findings by applying the IACS diagnostic criteriae.

In the Subjects and Methods section we need to know more about the Site: which community/communities are involved and their populations, or if census data are not available, at least an estimate thereof. Number of health clinics? Schools, boarding schools? The sentence beginning on line 90: “The district shares boundaries ..........” should be left out.

Line 94: are “structures” the same as houses?

The data collection team need to be better presented in a paragraph of its own. Criteria for their recruitment? How many were they? Were they especially trained for the task?

Samples from two populations were looked for, community members via house-to-house visits (within which local boundaries?) and school children via a random selection. What is meant with “.. all persons present ..” (line 101)? Present where? Some ideas of what constitutes the numerators are needed.

The REDCap questionnaire needs a better description and at least a reference. The standardised body examination was made as “in a previous study (7)”. The authors could have been a bit more detailed here and for that used as a good example that reference, a recent school study from the Solomon Islands (Osti MH et al. BMC Infect Dis. 2019): “Impetigo was diagnosed on the presence of papules, pustules or ulcerative lesions with associated erythema, crusting or pus. Skin examination consisted of assessment of exposed areas: the feet and legs to the thighs, hands to the upper arms, neck, face and scalp. Students were in school uniform which consisted of above-knee shorts and above-elbow shirts or dresses. Shoes were removed prior to examination.” Did the examination procedure differ between school-children and adults?

The assessment of the severity of scabies and impetigo merits some few comments than just a reference. Earlier treatment, how long ago was that asked for?

Statistical methods should be dealt with under its own heading. You have evidently used descriptive statistics with proportions. How do you deal with the uncertainties? With which methods did you make your analyses (SPSS/IBM does not do your thinking....). In the Results section, line 130, you introduce the acronym IQR without an earlier explanation under the Statistical methods. Your numbers in the different categories are small and the p-values have to be used with caution. Why are confidence intervals not used? Line 156: “The participants ....similar ..... scabies severity compared to the previously untreated participants (p=0.063).” This statement is not correct. You cannot judge similarity out of a p-value.

Line 120: Comparisons are made between proportions in subgroups. By “Linear-by-linear analysis” is probably meant an analysis for linear trend in categorical data.

The Results section is difficult to follow due to the lack of the information I have pointed out above. Does “Missing” in Table 1 mean ‘Missing information’ or ‘dropped out’?

Figure 2. “.... some participants.” = Four individuals, children, adults?

Table 2. The * is missing in the table.

Discussion and conclusion.

Line 170: “Here, we report......”. Already said, could be removed. As well as line 179: “Published data....”.

Line 171 onwards: When comparisons with other studies outside Ghana are made, it is important to comment on how these studies were made and whether comparisons are really valid.

Line 92: Are kidney diseases and rheumatic heart disease strikingly common in Ghana, or in the northern parts of Ghana?

The reference list contains 26 items and is up-to-date and comprehensive.

Reviewer #2: This is an important study on scabies in Ghana as there is limited data on scabies prevalence in much of West Africa but it requires substantial work to improve the description of the study methodology and the reporting of the results. 

1) In the abstract you say that scabies can be associated with RhD/RhF; i think the wording in the introduction where you say 'possibly' is more accurate and the abstract should be amended to be in line with the main text.

2) Line 65 - you say that fomites may play a role; this statment is relatively controversial as many would argue that fomites play a minimal role in transmission especially in LMIC/highly endemic settings. At a minimum this statement will need a reference.

3) Im not sure that line 76 adds anything - scabies is found/endemic everywhere, and having referenced 2 studies confirming scabies in Ghana I am not sure what is added by saying that scabies is endemic in Ghana. 

4) Line 79 - please provide more details. Was treatment limited strictly to cases or also their contacts? What treatment was provided by the municipal health authorities. 

5) Methods - the sampling frame needs a clearer description. How many communities were visited? What proportion of houses were visited? 

6) Line 103/4 I think you mean was recorded in a REDCAP database rather than obtained as presumably information was obtained by history/examination. 

7) Critically important is to understand is who conducted the examinations, what training they had received and how you confirmed they could correctly identify lesion morphology and distribution in line with the IACS criteria. This is not currently described and must be added. 

8) I presume children did not provide written informed consent but instead this was obtained from a parent/guardian. Please clarify. 

9) In the results and in line with the comment about sampling frame it is very unclear what % of individuals in the 4 communities were enrolled. Was it all residents? If not were there any systematic issues in sampling (i note for example that 61.5% of participants were female which suggests systematic bias in to study inclusion). 

10) The use of the IACS criteria needs revising. The authors state LINE 137 that a large proportion of patients had burros but then these individuals should be classified as B1 not B3. Please revise the methods and results accordingly. 

11) It would be useful to show some association data for scabies and impetigo as there is a lack of clarity on whether the strong relationship shown in the Pacific is seen elsewhere. This could be for example by including an odds ratio for impetigo amongst those with scabies and/or calculating the population attributable fraction of impetigo due to scabies.

12) It would be useful also to show scabies (and impetigo) prevalence stratified by key variables such as age and gender because rates often vary markedly by age group; given the non representative sampling methodology this is important to understand.

13) I would be careful comparing the prevalence found in this study, which deliberately targeted an area where there was high reporting of scabies cases, with representative sampling data. The limitations of these comparisons should be noted.

14) New results are presented in the discussion (see for example line 197-198) - please avoid this by including relevant information in the results section. 

15) Line 208 ivermectin doesnt need a capital letter

16) Paragraph beginning line 203; this section should clearly discuss the already existing evidence that treatment of index cases alone is associated with much higher rates of reinfestation; this is why (even outside the context of MDA) almost all guidelines recommend treating the whole house/close contacts.

Reviewer #3: (No Response)

PLOS authors have the option to publish the peer review history of their article (what does this mean?). If published, this will include your full peer review and any attached files.

Reviewer #1: No

Reviewer #2: No

Reviewer #3: Yes: Daniel K. Yeoh
---

## [Decision Letter · Decision Letter 1]

20 Jul 2020

Dear Dr Amoako,

Thank you very much for submitting your manuscript "A scabies outbreak in the North East Region of Ghana: the necessity for prompt intervention" for consideration at PLOS Neglected Tropical Diseases. As with all papers reviewed by the journal, your manuscript was reviewed by members of the editorial board and by several independent reviewers. The reviewers appreciated the attention to an important topic. Based on the reviews, we are likely to accept this manuscript for publication, providing that you modify the manuscript according to the review recommendations. 

Sincerely,

Alberto Novaes Ramos Jr, M.D., M.P.H., Ph.D.

Guest Editor

Aysegul Taylan Ozkan

Deputy Editor

Reviewer's Responses to Questions

**Key Review Criteria Required for Acceptance?**

**Methods**

-Are the objectives of the study clearly articulated with a clear testable hypothesis stated?

-Is the study design appropriate to address the stated objectives?

-Is the population clearly described and appropriate for the hypothesis being tested?

-Is the sample size sufficient to ensure adequate power to address the hypothesis being tested?

-Were correct statistical analysis used to support conclusions?

-Are there concerns about ethical or regulatory requirements being met?

Reviewer #1: The paper has been edited and is somewhat easier to follow, but there are still shortcomings of different kinds that need to be taken care of before it is publishable. 

Comments and suggestions that may be of help for the revision:

The Title is not covering the contents of the article. Suggested key words for the title are scabies, outbreak, prevalence, mass drug administration.

The Abstract needs to be rewritten. It is not giving a good and easy-to-read overview of the article.

 If the Introduction could begin like this “Scabies is an intensely pruritic skin disease caused by the mite....//persons (1).”, you would avoid the value-laden word “immense”. 

Line 50: A more correct way of using uppercase letters and the acronym: “making scabies one of the most common neglected tropical diseases (2). Recently, the WHO has classified scabies as a Neglected Tropical Disease (NTD) on the .....”

Line 55: instead of “acute and secondary complications”, “acute symptoms and secondary complications” is more logical.

Line 73: Better “.... estimated scabies worldwide to range from .....”.

Line 84: A clear stated aim/objectives is still missing. Since you conclude things at the end you must have had some intentions with your writing up of the data!

Line 99: The acronym IACS has not been explained before in the bread text and should be made here and not in line 124.

Line 104: The REDcap questionnaire needs a reference!

Line 107: “..... exposed regions of the skin was done in a previous study from Solomon Islands (10). Briefly, the skin examination.......”

Line 115: A table with the IACS criteria would be helpful here. E.g. with the text from lines174-180, given together with the text from the paragraph in line 123-130.

Line 138: The standard protocol in Ghana needs a reference.

Line 144-145: Don’t introduce an acronym for the Mann-Whitney test. It is not needed. The relative risk (RR) of .....//. .... calculated with 95% confidence interval (CI). Level of significance?

Line 152: “...<18 years provided assent for their...”. A bit unclear. Assent from whom?

Line 164: * Other instead of “other”.

Line 190: The statement here is still wrong!! You cannot write “similar scabies severity”! You cannot judge similarity out of a p-value. You do not need MWU here, just the p-value.

Line 193: .... treated for scabies were free...... Remove the comma.

Line 195: remove (44/260). The numbers are in the tables and the percentage will suffice here.

Line 196-197: 119 participants...... The sentence is a bit elliptic. Should also begin with “One-hundred-and-nineteen” to be consequent with what is done in Line 231: Eighty three......

Line 221-222: Scabies predominantly affects ...... Already said, does not need to be repeated.

Line 227-229: Remove sentence since it is already said and you have not studied the occurrence of the medical complications.

Line 252: Better “This is similar to the epidemiology of other NTDs of the skin like Buruli ulcer (27,28) and Yaws (29).”.

Line 257: Better “Treatment options for scabies in Ghana include......”

Line 303: Do you mean the prevalence of secondary infected scabies? Unclear sentence.

Line 325: Better “The examined students come from .......”

Line 328-331: Give these sentences a heading of its own, “Conclusion”

Reviewer #2: (No Response)

Reviewer #3: (No Response)

**Results**

-Does the analysis presented match the analysis plan?

-Are the results clearly and completely presented?

-Are the figures (Tables, Images) of sufficient quality for clarity?

Reviewer #1: Please, see comments under Methods!

Reviewer #2: (No Response)

Reviewer #3: (No Response)

**Conclusions**

-Are the conclusions supported by the data presented?

-Are the limitations of analysis clearly described?

-Do the authors discuss how these data can be helpful to advance our understanding of the topic under study?

-Is public health relevance addressed?

Reviewer #1: Please, see comments under Methods!

Reviewer #2: (No Response)

Reviewer #3: (No Response)

**Editorial and Data Presentation Modifications?**

Reviewer #1: Please, see comments under Methods!

Reviewer #2: (No Response)

Reviewer #3: (No Response)

**Summary and General Comments**

Reviewer #1: Please, see comments under Methods!

Reviewer #2: I am satisified the comments have been addressed

Reviewer #3: Well done – a much improved manuscript – minor comments below

Abstract

- Specify study design - cross sectional study to assess prevalence

- tighten paragraph on methods line 11-18 – could be shortened to 1-2 sentences. (important to mention skin assessment and IACS criteria here but less other details required)

- specify timing of treatment (recent treatment for scabies?) / exposure (how many weeks? (or could use “recent”) – may be easier to say “97% of participants reported a recent scabies contact”

- temper the conclusion re MDA – this study doesn’t demonstrate anything about efficacy of MDA – perhaps “Alternative strategies such as MDA may be required to contain outbreaks in such settings” or something regarding measuring efficacy

- revise sentence on MDA in Author’s summary section (as per comment above)

Methods

- as per comment in abstract – specify type of study (?prospective cross-sectional)

Results

- quantivy “much less” line 194 ?comparison done

Discussion

PLOS authors have the option to publish the peer review history of their article (what does this mean?). If published, this will include your full peer review and any attached files.

Reviewer #1: No

Reviewer #2: No

Reviewer #3: No
---

## [Decision Letter · Decision Letter 2]

17 Aug 2020

Dear Dr Amoako,

Thank you very much for submitting your manuscript "A scabies outbreak in the North East Region of Ghana: the necessity for prompt intervention" for consideration at PLOS Neglected Tropical Diseases. As with all papers reviewed by the journal, your manuscript was reviewed by members of the editorial board and by several independent reviewers. The reviewers appreciated the attention to an important topic. Based on the reviews, we are likely to accept this manuscript for publication, providing that you modify the manuscript according to the review recommendations. 

Sincerely,

Alberto Novaes Ramos Jr, M.D., M.P.H., Ph.D.

Guest Editor

Aysegul Taylan Ozkan

Deputy Editor

Reviewer's Responses to Questions

**Key Review Criteria Required for Acceptance?**

**Methods**

-Are the objectives of the study clearly articulated with a clear testable hypothesis stated?

-Is the study design appropriate to address the stated objectives?

-Is the population clearly described and appropriate for the hypothesis being tested?

-Is the sample size sufficient to ensure adequate power to address the hypothesis being tested?

-Were correct statistical analysis used to support conclusions?

-Are there concerns about ethical or regulatory requirements being met?

Reviewer #1: See below.

Reviewer #2: (No Response)

Reviewer #3: no concerns

**Results**

-Does the analysis presented match the analysis plan?

-Are the results clearly and completely presented?

-Are the figures (Tables, Images) of sufficient quality for clarity?

Reviewer #1: See below.

Reviewer #2: (No Response)

Reviewer #3: Results

- would be useful to include comparison of diagnostic classification between treated and non-treated groups - it looks like there is a difference between those with B3 and B1 diagnoses respectively (is this statistically significant?)

- if significant, this raises whether those with B3 diagnoses in the treated group may have resolving scabies (rather than re-infection) - worth including in the discussion as below

**Conclusions**

-Are the conclusions supported by the data presented?

-Are the limitations of analysis clearly described?

-Do the authors discuss how these data can be helpful to advance our understanding of the topic under study?

-Is public health relevance addressed?

Reviewer #1: See below.

Reviewer #2: (No Response)

Reviewer #3: Discussion 

- line 275 - although resolution of symptoms may take up to 4 weeks (https://www.nejm.org/doi/10.1056/NEJMcp052784)

- line 280 - revise this statement to consider limitations of timing of review post intervention

- perhaps something along the lines of:

"It is possible that some previously treated patients with an IACS classification of B3 may have resolving scabies rather than re-infection; the lower prevalence of burrows in the treated group compared with those not treated would support this possibility. However, given the timing of reassessment >2-3 weeks after, resolution of symptoms in the majority of treated patients would be expected (reference from benzoate trial by chisodow - https://www.who.int/bulletin/volumes/87/6/08-052308/en/)

**Editorial and Data Presentation Modifications?**

Reviewer #1: See below.

Reviewer #2: (No Response)

Reviewer #3: no concerns - minor revisions

**Summary and General Comments**

Reviewer #1: Unfortunately, there are still a lot to be criticized even after the second revision. My recommendation is that you re-write the paper and then resubmit it. An analysis of the scabies outbreak in Ghana is of public health interest but needs a more stringent presentation.

Some suggestions:

The Abstract needs to be rewritten, preferably after the bread text has been finished.

Lines 7-8: Is “we” = “a medical team” = “an assessment team”.

According to the text in lines 8-11 your methodology seems to be both quantitative (prevalence data and impact of earlier treatment) and qualitative ( ..”assess the practicality of using IACS....”).

Line 13: “...a prospective cross sectional study..”. The designation is not according to correct use of epidemiological concepts.

Line 50-51: “....making scabies one of the most common neglected tropical diseases”. Remove “neglected”.

Line 52: For syntactic reasons remove “on the basis of emerging evidence //... mass drug administration (MDA) and”.

Line 63: S. pyogenes instead of Streptococcus pyogenes, which is already introduced in line 61.

Line 71: remove “described as a disease of poverty and overcrowding and has been”.

Line 74-76. The sentence is too elliptic.

Line 84-89. Be more clear and precise about the aims and objectives. You are presenting prevalence data from a scabies outbreak, essentially a descriptive approach. Then you are comparing groups by age, sex, occupation, earlier treatment and existence of impetigo. This is the quantitative approach. But you are also judging whether IACS was a feasible tool, a qualitative approach.

Line 91: please see comment on line 13.

Line 95: The acronym SHS is not needed.

Line 98. The assessment team needs to be better presented. Is it the same as “we”, “the medical team” etc.?

Line 101. The acronym IACS is already introduced in line 85.

Line 103: What was randomly selected? Schools, classes, students?

Line 104: How many communities should be stated here and not just later in the text.

Line 107: The The REDCap reference leads –undated- to a Dutch website, but not to the questionnaire that has been used. A web-based program, that structures your data, is OK, but it does not tell us about how you constructed your questionnaire. The reader must be able to follow an ‘audit trail’ to make up her/his mind of the trustworthiness of the study.

It is of interest to know how many independent variables were used for the analysis.

Results: Use integers for the percentages, but consequently, throughout the entire bread text. Likewise numbers <10 are usually rendered by letters.

Line 165: “2018 IACS criteria”. Why 2018 here and not in the beginning?

Line 169: “crusted scabies” has not been mentioned before.

Line 183: Better to write ‘There was no statistically significant difference in the severity of scabies between the participants, who were treated by ? in the past months and the untreated participants (p=0.068).

Line 185: Remove “Impetigo seemed less common .....”. According to your testing, it was not less common! Why are you doing the statistical testing?

Discussion: Must be shortened and more focussed. There are too many repeats. The first sentence could be left out.

Discuss your findings and compare them to other prevalence studies, but don’t write too much about other treatments. It goes too far and beyond the scope of your study.

Line 220 -222 could be left out.

Line 223. CI for RR of impetigo...? . “...... implying .//..without scabies” could be left out.

Line 238: ..: “this will be important..... That sentence could be left out.

Line 280-281: The sentence is a bit cryptic.

Line 290. The acronym MDA is already presented in line 53.

Line 298: The word “tremendously” is too emotional to use in a scientific text.

Line 311: This sentence belongs to the Introduction and the Method sections.

Line 322: Remove “useful” and let the reader decide whether they are useful or not. The data could form a basis for future research.

References:

If a website is referred to, date for your reading it is needed. E.g. Ref 22.

Reviewer #2: I am satisfied the authors have made the requested amendments.

Reviewer #3: Thank you for the reviewed manuscript - just 2 minor comments regarding assessment in the treated population as outlined above

PLOS authors have the option to publish the peer review history of their article (what does this mean?). If published, this will include your full peer review and any attached files.

Reviewer #1: No

Reviewer #2: No

Reviewer #3: No
---

## [Decision Letter · Decision Letter 3]

3 Oct 2020

Dear Dr Amoako,

Thank you very much for submitting your manuscript "A scabies outbreak in the North East Region of Ghana: the necessity for prompt intervention" for consideration at PLOS Neglected Tropical Diseases. As with all papers reviewed by the journal, your manuscript was reviewed by members of the editorial board and by several independent reviewers. The reviewers appreciated the attention to an important topic. Based on the reviews, we are likely to accept this manuscript for publication, providing that you modify the manuscript according to the review recommendations. 

Sincerely,

Alberto Novaes Ramos Jr, M.D., M.P.H., Ph.D.

Guest Editor

Aysegul Taylan Ozkan

Deputy Editor

Reviewer's Responses to Questions

**Key Review Criteria Required for Acceptance?**

**Methods**

-Are the objectives of the study clearly articulated with a clear testable hypothesis stated?

-Is the study design appropriate to address the stated objectives?

-Is the population clearly described and appropriate for the hypothesis being tested?

-Is the sample size sufficient to ensure adequate power to address the hypothesis being tested?

-Were correct statistical analysis used to support conclusions?

-Are there concerns about ethical or regulatory requirements being met?

Reviewer #1: (No Response)

Reviewer #2: (No Response)

Reviewer #3: no concerns

**Results**

-Does the analysis presented match the analysis plan?

-Are the results clearly and completely presented?

-Are the figures (Tables, Images) of sufficient quality for clarity?

Reviewer #1: (No Response)

Reviewer #2: (No Response)

Reviewer #3: no concernsv

**Conclusions**

-Are the conclusions supported by the data presented?

-Are the limitations of analysis clearly described?

-Do the authors discuss how these data can be helpful to advance our understanding of the topic under study?

-Is public health relevance addressed?

Reviewer #1: (No Response)

Reviewer #2: (No Response)

Reviewer #3: no concerns

**Editorial and Data Presentation Modifications?**

Reviewer #1: (No Response)

Reviewer #2: (No Response)

Reviewer #3: no concerns

**Summary and General Comments**

Reviewer #1: The manuscript has benefitted from the revisions, but I still have some comments:

Line 84: “assess the practicality” implies a scientific approach. Your assessment, however, seems to be based on the subjective judgements by the team members.

Line 106: The REDCap mobile app is not wellknown enough to be presented like that. I strongly recommend that the questionnaire is added as a Supplementary file to the manuscript.

Line 226: “MDA” is mentioned for the first time in the bread text and therefore has to be explained, i.e. “ mass drug administration (MDA)”.

Reviewer #2: I believe the authors have responded reasonably to the comments from the peer reviewers

Reviewer #3: I am satisfied with this revised version of the manuscript

PLOS authors have the option to publish the peer review history of their article (what does this mean?). If published, this will include your full peer review and any attached files.

Reviewer #1: No

Reviewer #2: No

Reviewer #3: No
---

## [Decision Letter · Decision Letter 4]

19 Oct 2020

Dear Dr Amoako,

We are pleased to inform you that your manuscript 'A scabies outbreak in the North East Region of Ghana: the necessity for prompt intervention' has been provisionally accepted for publication in PLOS Neglected Tropical Diseases.

Best regards,

Alberto Novaes Ramos Jr, M.D., M.P.H., Ph.D.

Guest Editor

Aysegul Taylan Ozkan

Deputy Editor

Reviewer's Responses to Questions

**Key Review Criteria Required for Acceptance?**

**Methods**

-Are the objectives of the study clearly articulated with a clear testable hypothesis stated?

-Is the study design appropriate to address the stated objectives?

-Is the population clearly described and appropriate for the hypothesis being tested?

-Is the sample size sufficient to ensure adequate power to address the hypothesis being tested?

-Were correct statistical analysis used to support conclusions?

-Are there concerns about ethical or regulatory requirements being met?

Reviewer #2: (No Response)

Reviewer #3: (No Response)

**Results**

-Does the analysis presented match the analysis plan?

-Are the results clearly and completely presented?

-Are the figures (Tables, Images) of sufficient quality for clarity?

Reviewer #2: (No Response)

Reviewer #3: (No Response)

**Conclusions**

-Are the conclusions supported by the data presented?

-Are the limitations of analysis clearly described?

-Do the authors discuss how these data can be helpful to advance our understanding of the topic under study?

-Is public health relevance addressed?

Reviewer #2: (No Response)

Reviewer #3: (No Response)

**Editorial and Data Presentation Modifications?**

Reviewer #2: (No Response)

Reviewer #3: (No Response)

**Summary and General Comments**

Reviewer #2: The authors have responded appropriately

Reviewer #3: (No Response)

PLOS authors have the option to publish the peer review history of their article (what does this mean?). If published, this will include your full peer review and any attached files.

Reviewer #2: No

Reviewer #3: **Yes: **Daniel K Yeoh

---

## [Editor Report · Acceptance letter]

9 Dec 2020

Dear Dr Amoako,

We are delighted to inform you that your manuscript, "A scabies outbreak in the North East Region of Ghana: the necessity for prompt intervention," has been formally accepted for publication in PLOS Neglected Tropical Diseases.

Best regards,

Shaden Kamhawi

co-Editor-in-Chief

Paul Brindley

co-Editor-in-Chief
